# Never Seen Before: Benchmarking Genuine Zero-Shot Composed Image Retrieval with Consistent Video-Sourced Datasets

## Abstract

Zero-Shot Composed Image Retrieval (ZS-CIR) aims to retrieve a target image based on a query composed of a reference image and a relative caption without training samples. Existing ZS-CIR datasets often suffer from complete irrelevance between reference and target images due to noisy image sources, and do not achieve a true zero-shot scenario as they use public image datasets that models like CLIP have been trained on. To tackle these challenges, we introduce ZeroSight, a novel benchmark for ZS-CIR. It includes a dataset with consistent reference-target pairs sourced from videos, a data construction pipeline, and evaluation methods that consider the ranking of multiple positive and negative target images. We ensure visually and semantically consistent reference-target pairs by extracting frames from a single video and generating relative captions using LLM-assisted methods. To ensure a true zero-shot scenario, we use video data published after March 31, 2022, ensuring it was not included in CLIP's pre-training data. Additionally, we propose a training-free MLLM-driven method, SC4CIR (Symmetric Consistency for CIR), which can effectively identify hard negative targets through 3 symmetric consistency checks. This method is plug-and-play, seamlessly integrating with various CIR methods and significantly improving performance. Our experimental results from 27 methods reveal that current ZS-CIR datasets and evaluation metrics result in inflated retrieval performance, exaggerating the capabilities of CIR methods. Our benchmark and models can be accessed at `https://anonymous.4open.science/r/ZeroSight-CFE1`.

## 1 Introduction

Composed Image Retrieval (CIR) Zhang et al. (2021); Baldrati et al. (2022b); Wen et al. (2023) extends traditional cross-modal retrieval tasks by integrating textual descriptions into the image search process, which allows users to search for images based on visual attributes while specifying particular modifications to the query image. Traditional CIR methods require labor-intensive creation of annotated triplets consisting of a reference image, modification text, and target image for training. To mitigate this issue, the recent Zero-Shot CIR (ZS-CIR) task Saito et al. (2023); Baldrati et al. (2023); Tang et al. (2023) has been introduced, which aims to enhance the generalization capabilities of CIR methods without relying on annotated triplets, while still maintaining high retrieval accuracy.

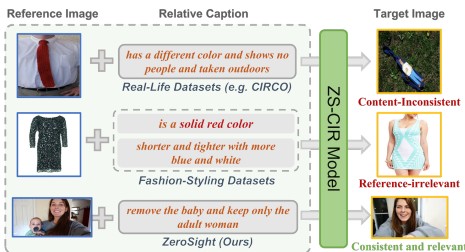

Figure 1: Comparison of CIR among existing datasets. For real-life datasets, the image pairs are semantically and visually inconsistent, while for fashion datasets, the caption is noisy and the target is irrelevant to the reference (tabulated in App. J). The image pairs of ZeroSight come from the same video and are both consistent and relevant.

Rich datasets Baldrati et al. (2023); Liu et al. (2021); Vaze et al. (2023); Wu et al. (2021); Levy et al. (2024) for CIR integrate compositional learning Kim et al. (2021); Hou et al. (2020); Sun et al. (2025); Tian et al.; Ventura et al. (2024); Liu et al. (2024); Xu et al. (2024) with image retrieval. This fusion

creates a range of challenging tasks that have been widely applied in fashion styling Wu et al. (2021) and conditional search Baldrati et al. (2022b); Vaze et al. (2023). For instance, FashionIQ Wu et al. (2021) focuses on image retrieval in fashion styling, while GeneCIS Vaze et al. (2023) emphasizes the ability of models to adapt to various similarity conditions. However, these datasets typically have weakly related reference images and relatively noisy text descriptions, as shown in Figure 1. CIRR Liu et al. (2021) focuses on real-life images, derived from NLVR2 Suhr et al. (2017). However, it suffers from two main issues. First, the dataset contains several false negatives, which can lead to inaccurate evaluations. Second, the queries often do not consider the visual content of the reference image, making the task solvable with standard text-to-image techniques. CIRCO Baldrati et al. (2023), derived from MS COCO Lin et al. (2014), addresses these issues by a strategy that leverages CIR methods to ease the annotation process of multiple ground truths. These datasets provide a diverse set of images, making them valuable resources for advancing research in CIR.

However, previous CIR datasets mistakenly treat ZS-CIR as an abstract retrieval task, where the visual and semantic aspects of the target image are not strictly determined by the query. The abstract semantic composition causes inconsistencies between reference and target images because these datasets come from noisy image datasets, where there is a natural inconsistency between images of the same abstract concept. For example, CIRR Liu et al. (2021) constructs similar image sets from NLVR2 Suhr et al. (2017) using a clustering-like method and creates reference-target image pairs within these sets. However, NLVR2 is crowdsourced from the web, resulting in gaps even among similar images, which only share abstract conceptual similarities and can be tenuously linked through relative captions. For instance, in Figure 2, the content of the image pairs sampled from a cluster is completely unrelated, except for the semantic 'tie'.

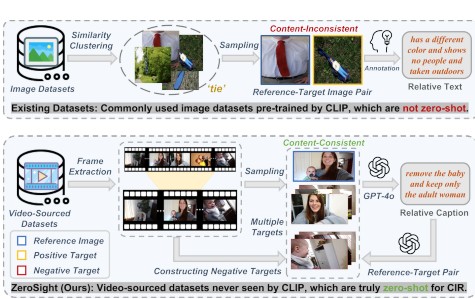

Figure 2: Comparison of existing ZS-CIR dataset construction pipelines. The current ZS-CIR datasets face two difficulties: (1) There are natural inconsistencies in image pairs constructed from image datasets; (2) The commonly used public image datasets are pre-trained by CLIP. ZeroSight solves both of them by constructing consistent image pairs using video datasets from after 2022.

In fact, we expect the reference and target images to remain as consistent as possible in aspects not mentioned in the relative caption. From the preceding discussion, we aim to address **Challenge 1**: How to construct visually and semantically consistent reference-target image pairs in CIR datasets?

Furthermore, most existing datasets overlook a critical point in the zero-shot CIR task: the model should not have access to test image data. However, the image pairs in existing datasets mostly come from commonly used public image datasets such as MS COCO Lin et al. (2014) and ImageNet Deng et al. (2009). Additionally, most approaches Cohen et al. (2022a); Saito et al. (2023); Baldrati et al. (2023); Gu et al. (2023); Karthik et al. (2023); Yang et al. (2024b;a); Bao et al. (2025); Wang et al. for ZS-CIR leverage the cross-modal alignment capabilities of large-scale pre-trained Vision-Language Models (VLMs) (e.g., CLIP Radford et al. (2021)) to extract aligned features from images and text for retrieval, which uses a large number of datasets scraped from the internet, including LAION-2B Schuhmann et al. (2022), DataComp-1B Gadre et al. (2024), WIT Srinivasan et al. (2021), WebLI Chen et al. (2022), and DFN-5B Fang et al. (2023). These datasets cover many commonly used public datasets. For example, CIRCO Baldrati et al. (2023) and GeneCIS Vaze et al. (2023) focus on real-life images, constructing image datasets from MS COCO Lin et al. (2014), which has been used to train CLIP. Therefore, the existing datasets do not achieve a true zero-shot scenario, inflating results due to the negative impact of pre-trained CLIP (see app. L). Inspired by that, we have to tackle **Challenge 2**: How to use data that CLIP has never been trained on to construct a truly zero-shot composed image retrieval dataset?

To address these challenges, we introduce **ZeroSight**, a benchmark with consistent reference-target pairs from video-sourced datasets for ZS-CIR. For **Challenge 1**, we extract frame images from a single video to construct visually and semantically consistent image pairs. We start by collecting high-quality video datasets, then refine and select non-redundant, distinct reference images. For each reference image, we filter and choose multiple content-consistent target images from the same video. Finally, we use LLM-assisted generation to produce relative captions for the consistent image pairs. For **Challenge 2**, we utilize video data that CLIP has never been trained on to construct a completely zero-shot CIR dataset. To ensure this, we incorporate video datasets from after March 31, 2022, the

Table 1: The comparison of different ZS-CIR datasets. **#Index**: the size of the retrieval pool shared by all queries. **Specific**: whether the dataset is specific to CIR. **Open-Domain**: whether the query types are diverse. **Multi-Target**: whether there are multiple ground truths. **Hard-Negative**: whether to provide hard negative target images. **Zero-Shot**: whether the images are definitely not pre-trained with CLIP.

| Dataset | #Query | #Index | Specific | Open-Domain | Multi-Target | Hard-Negative | Zero-Shot | Image Source |
|---|---|---|---|---|---|---|---|---|
| CIRR Liu et al. (2021) | 4,148 | 2,316 | ✔ | ✔ | ✗ | ✗ | ✗ | NLVR2 |
| CIRCO Baldrati et al. (2023) | 800 | 123,403 | ✔ | ✔ | ✔ | ✗ | ✗ | MS COCO |
| GeneCIS Vaze et al. (2023) | 2,008 | - | ✗ | ✔ | ✗ | ✗ | ✗ | MS COCO |
| FashionIQ Wu et al. (2021) | 2,005 | 5,179 | ✔ | ✗ | ✗ | ✗ | ✗ | Shopping Sites |
| **ZeroSight (Ours)** | **54,740** | **197,313** | ✔ | ✔ | ✔ | ✔ | ✔ | **Videos** |

release date of the most recent CLIP dataset LAION-2B Schuhmann et al. (2022), ensuring that they have not been included in the pre-training data of CLIP.

Additionally, to exclude hard negative targets that are highly similar to the target in retrieval results, we propose a training-free MLLM-driven ZS-CIR method, **SC4CIR** (Symmetric Consistency for CIR). This method uses a forward retrieval and two reverse processes to identify hard negative targets through consistency checks. It is plug-and-play, seamlessly integrating with various CIR methods and significantly improving performance, especially enhancing LLM-based ZS-CIR methods by 9.12%.

In summary, our contributions are as follows:

- We propose **ZeroSight**, a novel benchmark with consistent reference-target pairs from video-sourced datasets for ZS-CIR. It encompasses a CIR dataset, a data construction pipeline, and a set of evaluation methods designed to thoroughly assess current CIR methods considering the ranking of multiple positive and negative target images.

- We design a multi-stage LLM-assisted dataset construction pipeline to create a truly zero-shot CIR dataset from 12,048 diverse videos, comprising 197,313 candidate images in the retrieval pool and 54,740 queries. On average, each query features 5.16 positive target images and 10.89 negative target images. This is the first dataset to include multiple ground truths and hard negative targets.

- We introduce a training-free MLLM-driven method called **SC4CIR** (Symmetric Consistency for CIR), designed to effectively identify hard negative targets through 3 symmetric consistency checks. This method is plug-and-play, seamlessly integrating with various CIR methods and significantly improving performance.

- We release the leaderboard for ZS-CIR on ZeroSight. Extensive experiments conducted on it demonstrate that current CLIP-based CIR methods show inflated results on existing ZS-CIR datasets, exaggerating their capabilities.

## 2 RELATED WORK

**Datasets for Composed Image Retrieval**  Composed Image Retrieval (CIR) task Vo et al. (2019); Chen et al. (2020a); Gu et al. (2021); Lee et al. (2021); Liu et al. (2023), focusing on retrieving a target image based on a query of a reference image and a relative caption, has gained significant attention in recent years. Rich datasets Liu et al. (2021); Vaze et al. (2023); Wu et al. (2021) for CIR integrate compositional learning Kim et al. (2021); Hou et al. (2020); Sun et al. (2025) with image retrieval, forming a series of demanding tasks, which have been widely applied in fashion styling Wu et al. (2021) and conditional search Baldrati et al. (2022b); Vaze et al. (2023). For instance, FashionIQ Wu et al. (2021) focuses on image retrieval in fashion styling, while GeneCIS Vaze et al. (2023) emphasizes the ability of models to adapt to various similarity conditions in conditional search. The CIRCO Baldrati et al. (2023) and CIRR Liu et al. (2021) datasets, which are derived from MS COCO Lin et al. (2014) and NLVR2 Suhr et al. (2017), focus on common objects and real-life images. However, they treat ZS-CIR as an abstract retrieval task, leading to inconsistencies between reference and target images. To overcome it, we extract frame images from high-quality videos to create visually and semantically consistent image pairs.

**Vision-Language Models for ZS-CIR**  The popularity of the pre-trained BERT Devlin et al. (2018) model has sparked interest in developing pre-trained Vision-Language Models (VLMs),

Figure 3: **Overview of the proposed ZeroSight framework.** We design a multi-stage LLM-assisted dataset construction pipeline to create a truly zero-shot CIR dataset from diverse videos, comprising 197,313 candidate images in the retrieval pool and 54,740 queries. This is the first dataset to include multiple ground truths and hard negative target images. In the pipeline, each step is presented *sequentially* using numerical order. And three different screening criteria are involved, each indicated by *a distinct background color*.

including Chen et al. (2020b); Li et al. (2019; 2020); Lu et al. (2019); Tan & Bansal (2019), aiming to create Transformer-based Vaswani et al. (2017) models trained on large-scale image-text triplets to produce vision-and-language representations. For CIR, to map images and text into a shared embedding space, many methods harness large pre-trained VLMs, such as CLIP Radford et al. (2021), as the backbone for feature extraction. These models Baldrati et al. (2022a); Han et al. (2023); Karthik et al. (2023) have recently gained popularity due to their exceptional ability to handle multi-modal data. However, most existing ZS-CIR datasets fail to ensure a true zero-shot scenario as they use public image datasets that CLIP have been trained on. To address this, we construct a zero-shot CIR dataset using video data sourced after March 31, 2022, ensuring it has not been included in CLIP's pre-training data.

# 3 DATASET: ZEROSIGHT

We develop a comprehensive data collection pipeline to construct a high-quality ZS-CIR dataset, tailored for annotating consistent reference-target pairs, as shown in Figure 3. The comparison of our proposed ZeroSight with other ZS-CIR datasets is shown in Table 1.

## 3.1 DATA COLLECTION

To ensure that our dataset truly aligns with the definition of ZS-CIR, we do not utilize any existing image datasets or collect images from various sources. Instead, we pioneer the use of video datasets as the initial data sources. Then we extract frames from each video to use as images for our dataset. This approach leverages the inherent narrative continuity of videos, ensuring that within a single video, there are pairs of frames that, while similar in content (such as people, objects, etc.), differ in appearance. To differentiate it from LAION-5B Schuhmann et al. (2022), we filter out videos that were published after March 31, 2022 from YT-Temporal-1B Zellers et al. (2022), ensuring that the video data are sufficiently recent. Additionally, to ensure the a rich variety of the images in our dataset, we further categorize the filtered videos. The final videos are divided into 12 main categories and 36 subcategories. Details about the distribution of video categories are included in App. D.

## 3.2 DATASET CONSTRUCTION PIPELINE

We propose a semi-automated dataset construction pipeline for generating multiple ground-truth reference-target image pairs with relative captions to construct a ZS-CIR dataset from videos. This pipeline comprises a multi-stage LLM-assisted generation process and a series of interleaved filtering processes, which takes about 3 months and involves over 30 professional annotators. Although our dataset is derived from a collection of frames extracted from diverse videos, when constructing the reference-target image pairs, we utilize only the frame images $F = \{f_i\}$ from a single video to ensure the consistency and validity of the constructed results. Detailed prompts are provided in App. O.

**Generating Reference Images**    To ensure the diversity of ZeroSight, the reference images are non-redundant and exhibit significant differences. Firstly, we divide video frames into equally spaced subsets $c_i$, each containing 10 frames, resulting in the set of equally spaced subsets $C = \{c_i | 1 \leq i \leq \lceil \frac{|F|}{10} \rceil \}$. Then, we employ MLLMs (such as GPT-4o Achiam et al. (2023)) to initially filter these subsets and select the candidate reference image $r_j$, as follows:

$$r_j = \begin{cases} \text{MLLM}(p_r^1, c_i) & \text{if } j = 1 \\ \text{MLLM}(p_r^2, c_i, r_{j-1}) & \text{if } j > 1 \end{cases}. \tag{1}$$

Thus, we obtain the set of candidate reference images $Rt = \{r_j | 1 \leq j \leq |C|\}$. Here, $p_r^1$ and $p_r^2$ are the prompts used to avoid similar images when initially selecting candidate reference images using MLLMs. To further ensure the relative independence of each selected image, we design additional steps. First, we use Vision Transformer (ViT) Dosovitskiy (2020) to remove visually over-consistent reference images, resulting in the set $Rv = \{Rv_i \in Rt | \text{ViT}(Rv_i, \alpha_1)\}$, which consists of candidate reference images that are relatively independent of each other on a visual level. Here, $\alpha_1 = 0.50$ is the upper limit of the visual similarity threshold for filtering. Then we use the CLIP model to remove semantically over-consistent reference images. The final result is a set of reference images that are relatively independent both visually and semantically, $R = \{Rs_i \in Rv | \text{CLIP}(Rs_i, \beta_1)\}$, where $\beta_1 = 0.80$ is the upper limit of the semantic similarity threshold for filtering.

**Generating Multiple Target Images**    After generating the set of reference images from a video, we construct multiple target images from the same video $F$ for each reference image. The goal is to select images that have a certain degree of similarity to the reference image but are not completely identical. To achieve this, we first use ViT to filter out $Sv_i = \{Sv_j^i \in F | \text{ViT}(Sv_j^i, Rs_i, \alpha_2, \alpha_3), Rs_i \in R\}$, the set of images that have a certain visual similarity to the $i$-th reference image $Rs_i$, where $\alpha_2 = 0.35$ and $\alpha_3 = 0.50$ are the lower and upper limits of visual similarity thresholds, respectively. Next, to further ensure a semantic connection between the filtered similar images and the reference image, we consider the semantic similarity of these images to the reference image, and use CLIP to further filter out $Ss_i = \{Ss_j^i \in Sv | \text{CLIP}(Ss_j^i, Rs_i, \beta_2, \beta_3), Rs_i \in R\}$, the set of images with a certain degree of semantic similarity to $Rs_i$, where $\beta_2 = 0.65$ and $\beta_3 = 0.75$ are respectively the lower and upper limits of semantic similarity thresholds. Finally, to ensure that the set of target images corresponding to each reference image does not contain visually redundant images, we use ViT again to filter out images in $Ss_i$ that are visually over-similar, as shown below:

$$St_i = \{St_j^i \in Ss_i | \text{ViT}(St_j^i, Ss_m^i, \theta), Ss_m^i \in Ss_i, m \neq j\}, \tag{2}$$

which represents the final set of candidate target images. Here $\theta = 0.85$ is the upper limit of the visual similarity threshold for filtering. Next, for the videos that are not used to generate $St = \{St_i | 1 \leq i \leq |R|\}$, we extract frame images at equal intervals of every 5 frames. These images are then combined with each $St$ to form the retrieval pool.

**Generating Relative Captions**    After generating reference image sets and candidate target image sets, we use MLLMs to generate candidate relative caption for each reference image and each candidate target image in its corresponding candidate set. Then we construct $Tt_i = \{t_j^i = \text{MLLMs}(Rs_i, St^j) | Rs_i \in R, St^j \in St_i\}$, a set of candidate relative captions for the $i$-th reference image $Rs_i$, where $St_i$ is the set of candidate target images corresponding to the $i$-th reference image $Rs_i$, $St^j$ is the $j$-th candidate target image in $St^i$, and $t_j^i$ represents the candidate relative caption generated for the $i$-th reference image and the $j$-th candidate target image in $St_i$. Subsequently, we use BERT Devlin et al. (2018) to calculate the textual similarity for all pairs of candidate relative captions within $Tt_i$, resulting in the set $Vx = \{v_x^y = \text{BERT}(x, y) | x, y \in Tt_i, y \neq x\}$, which represents the textual similarities calculated between the $x$-th candidate relative caption in $Tt_i$ and all other candidate relative captions in the same set. To obtain the final relative caption, we define the following function:

$$k(x) = max\{\epsilon | \frac{|\{v_x^y \in Vx | v_x^y \geq \epsilon\}|}{|Vx|} \geq \epsilon\}, \tag{3}$$

where $k$ ($0 \leq k \leq 1$) is a variable parameter. We then construct the set $T_i$ as follows:

$$Ts_i = \{t_j \in Tt_i | \text{BERT}(\underset{x \in Tt_i}{\arg\max} \, k(x), t_j) \geq \underset{x \in Tt_i}{\max} \, k(x)\}, \tag{4}$$

which represents the subset of $Tt_i$ used to generate the final relative caption. Then, we can generate the final target image set $Sp_i$ for the $i$-th reference image $Rs_i$ as follows:

$$Sp_i = \{p_m = St_j | St_j \in St, t_j^i \in Tt_i, t_j^i \in Ts_i\}, \tag{5}$$

which is the set of positive target images for the $i$-th reference image $Rs_i$. Conversely, the set of negative target images for the $i$-th reference image $Rs_i$ is is represented as $Sn_i = \{n_m \in St | n_m \notin Sp_i\}$. This unique characteristic distinguishes our dataset from others. Subsequently, we utilize LLMs (such as GPT-4) to generate a unified final relative caption $Tl_i = \text{LLM}(Ts_i, p_t)$ for $Rs_i$, along with the corresponding $Sp_i$ and $Sn_i$, where $p_t$ represents the prompt used to generate a text that summarizes the meaning of $t_j (t_j \in Ts_j)$. Then we obtain a triplet $(Rs_i, Tl_i, \langle Sp_i, Sn_i \rangle)$ for $Rs_i (Rs_i \in R)$ as the query structure. Finally, to ensure the quality of ZeroSight and MLLM annotations, we manually review and select appropriate queries. Additionally, we review and score MLLM from 4 dimensions, and finally use GPT-4o for pipeline construction, as detailed in App. E.

## 4 METHOD: SC4CIR

As mentioned above, existing datasets and methods do not account for hard negatives that are highly similar to the target. In our ZeroSight benchmark, each query includes an average of 10.89 hard negative images, increasing the retrieval difficulty. To address this, we propose a training-free MLLM-driven method, SC4CIR (Symmetric Consistency for CIR), which uses a forward retrieval and two reverse processes to identify hard negative targets through consistency checks. This method is plug-and-play, seamlessly integrating with various CIR methods and significantly improving performance, especially LLM-based CIR methods.

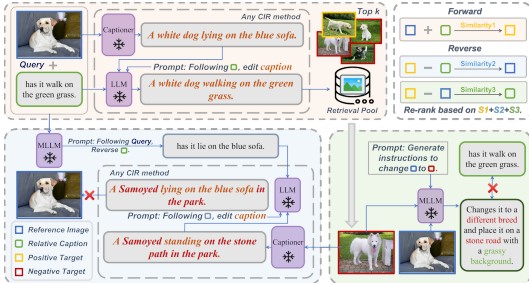

Figure 4: **Architecture of the proposed SC4CIR method.** Forward and bidirectional reverse retrieval flows are distinguished by color blocks.

### 4.1 SYMMETRIC CONSISTENCY CHECKING

As shown in Figure 4, most CIR methods mistakenly treat hard negative images as targets due to the ambiguity in the forward retrieval process. For example, in the forward process $I^r + T \rightarrow I^t$, where $I^r$ represents the reference image, $T$ represents the relative text and $I^t$ represents the retrieved candidate targets (which may be positive or negative images), CIReVL Karthik et al. (2024) successfully retrieves images that match the 'white dog' semantics. However, most of these images (e.g., 'Samoyed') do not match the reference image. Therefore, we introduce two symmetric reverse processes to verify the consistency of the results. For the top $N$ ($N = 30$) candidates retrieved in the forward process, we separately calculate the two similarities in the reverse process as follows:

**Reverse Process 1:** $I^t - T \rightarrow I^r$. We replace the subtraction operation with the generation of an instruction $T'$, which is the inverse of $T$, i.e., $I^t + T' \rightarrow I^r$. Since the reverse operation requires information from the reference image, we utilize an MLLM (GPT-4o) to generate $T' = \phi(I^r, T, p_1)$. Here, $\phi$ represents the MLLM used and $p_1$ is a prompt that directs the model to generate the reverse operation $T'$ based on $I^r$ and $T$. We then apply the same CIR method, using $I^t$ and $T'$ as inputs for retrieval. Finally, we calculate the similarity $S_2$ between retrieval results and the reference image $I^r$.

**Reverse Process 2:** $I^t - I^r \rightarrow T$. We achieve the subtraction operation by describing the differences between the reference image $I^r$ and the candidate target $I^t$. We use GPT-4o to generate a relative caption for the candidate target, $T'' = \phi(I^r, I^t, p_2)$, where $p_2$ is a prompt that guides the model to generate the instruction $T''$ that transforms $I^r$ into $I^t$. We then verify the consistency between the original relative caption $T$ and the generated relative caption $T''$ by calculating their similarity $S_3$.

### 4.2 POSITIVE-NEGATIVE RE-RANKING

For the top $N$ candidate targets in the retrieval results, the lower the similarity in the reverse process, the worse the symmetric consistency, and the more likely it is to be a negative target. In the forward

Table 2: Results of comparison among different ZS-CIR methods on ZeroSight.

| Backbone | Method | Training-free | mAP@k | | | | | PNR-mAP@k | | | | |
|---|---|---|---|---|---|---|---|---|---|---|---|---|
| | | | k=5 | k=10 | k=25 | k=50 | Avg. | k=5 | k=10 | k=25 | k=50 | Avg. |
| ViT-B/32 | CIReVL(ICLR'2024) | ✔ | 7.76 | 8.51 | 9.57 | 10.12 | 8.99 | 4.95 | 6.00 | 7.28 | 7.96 | 6.55 |
| | LDRE(SIGIR'2024) | ✔ | 9.53 | 10.46 | 11.79 | 12.47 | 11.06 | 6.01 | 7.33 | 8.93 | 9.77 | 8.01 |
| | SEIZE(MM'2024) | ✔ | 10.64 | 11.61 | 12.94 | 13.65 | 12.21 | 6.39 | 7.82 | 9.44 | 10.30 | 8.49 |
| | PALAVRA(ECCV'2022) | ✘ | 12.67 | 13.89 | 15.51 | 16.36 | 14.61 | 7.99 | 9.70 | 11.62 | 12.61 | 10.48 |
| | SEARLE-OTI(ICCV'2023) | ✘ | 21.39 | 24.17 | 26.45 | 27.27 | 24.82 | 19.01 | 22.40 | 25.19 | 26.19 | 23.20 |
| | SEARLE(ICCV'2023) | ✘ | 22.65 | 25.58 | 27.98 | 28.84 | 26.26 | 20.15 | 23.72 | 26.65 | 27.70 | 24.56 |
| | SEIZE+SC4CIR(Ours) | ✔ | 12.59 | 13.71 | 15.21 | 16.12 | 14.41 | 9.48 | 10.93 | 11.98 | 13.41 | 11.45 |
| ViT-L/14 | Captioning | ✔ | 5.86 | 6.68 | 7.74 | 8.32 | 7.15 | 3.84 | 4.94 | 6.22 | 6.94 | 5.49 |
| | Text-only | ✔ | 8.38 | 9.19 | 10.27 | 10.84 | 9.67 | 5.26 | 6.40 | 7.68 | 8.34 | 6.92 |
| | CIReVL(ICLR'2024) | ✔ | 9.45 | 10.40 | 11.74 | 12.42 | 11.00 | 5.95 | 7.29 | 8.91 | 9.74 | 7.97 |
| | Image-only | ✔ | 15.62 | 17.26 | 19.38 | 20.54 | 18.20 | 5.58 | 7.78 | 10.34 | 11.78 | 8.87 |
| | LDRE(SIGIR'2024) | ✔ | 12.60 | 14.00 | 15.62 | 16.44 | 14.67 | 8.01 | 9.89 | 11.87 | 12.87 | 10.66 |
| | SEIZE(MM'2024) | ✔ | 13.02 | 14.36 | 15.98 | 16.80 | 15.04 | 8.21 | 10.04 | 12.02 | 13.01 | 10.82 |
| | Pic2Word(CVPR'2023) | ✘ | 15.06 | 18.09 | 19.78 | 20.21 | 18.28 | 12.63 | 16.32 | 18.46 | 18.96 | 16.59 |
| | SEARLE-OTI(ICCV'2023) | ✘ | 24.93 | 30.68 | 33.89 | 34.71 | 31.05 | 20.32 | 27.33 | 31.39 | 32.33 | 27.84 |
| | SEARLE(ICCV'2023) | ✘ | 26.12 | 32.17 | 35.55 | 36.41 | 32.56 | 21.26 | 28.64 | 32.92 | 33.91 | 29.18 |
| | LinCIR(CVPR'2024) | ✘ | 28.13 | 34.48 | 38.03 | 38.93 | 34.89 | 23.02 | 30.77 | 35.27 | 36.31 | 31.34 |
| | SEIZE+SC4CIR(Ours) | ✔ | 15.13 | 16.56 | 18.03 | 18.91 | 17.16 | 11.41 | 13.28 | 15.16 | 16.13 | 14.00 |
| | LinCIR+SC4CIR(Ours) | ✘ | 29.24 | 35.63 | 39.36 | 36.01 | 35.06 | 25.13 | 32.96 | 37.35 | 38.50 | 33.49 |
| ViT-G/14 | CIReVL(ICLR'2024) | ✔ | 12.00 | 13.24 | 14.74 | 15.49 | 13.87 | 7.56 | 9.25 | 11.08 | 12.00 | 9.97 |
| | LDRE(SIGIR'2024) | ✔ | 15.98 | 17.64 | 19.64 | 20.65 | 18.48 | 10.04 | 12.30 | 14.75 | 15.97 | 13.27 |
| | SEIZE(MM'2024) | ✔ | 16.63 | 18.37 | 20.47 | 21.54 | 19.25 | 10.37 | 12.75 | 15.33 | 16.61 | 13.77 |
| | Pic2Word(CVPR'2023) | ✘ | 16.07 | 19.39 | 21.25 | 21.73 | 19.61 | 13.39 | 17.45 | 19.81 | 20.35 | 17.75 |
| | SEARLE(ICCV'2023) | ✘ | 31.46 | 37.95 | 41.72 | 41.83 | 38.23 | 23.14 | 31.73 | 35.30 | 36.58 | 31.69 |
| | LinCIR(CVPR'2024) | ✘ | 32.00 | 38.57 | 42.38 | 42.51 | 38.87 | 24.62 | 33.31 | 37.93 | 38.23 | 33.52 |
| | SEIZE+SC4CIR(Ours) | ✔ | 18.11 | 19.76 | 21.93 | 22.98 | 20.70 | 12.68 | 15.14 | 17.69 | 18.98 | 16.12 |
| | LinCIR+SC4CIR(Ours) | ✘ | 33.04 | 39.68 | 43.57 | 43.79 | 40.02 | 26.55 | 35.28 | 39.90 | 40.21 | 35.49 |

retrieval process, the similarity with $I^r$ is denoted as $S_1$, and in the reverse processes 1 and 2, the similarities obtained are $S_2$ and $S_3$, respectively. Based on these similarities, we calculate the overall similarity $S = S_1 + S_2 + S_3$ and re-rank the retrieval results accordingly to obtain the final retrieval results. This result is based on symmetric consistency checking, ensuring the reliability and accuracy.

## 5 EXPERIMENTS

### 5.1 EXPERIMENTAL SETUP

To effectively evaluate the performance of current both ZS-CIR and CIR methods, we meticulously select a diverse range of state-of-the-art methods, as detailed in App. C. For the two categories of methods, we design two distinct experimental setups within ZeroSight to rigorously assess the capabilities.

**Evaluation for ZS-CIR Methods.** In this setup, we evaluate the capabilities of ZS-CIR methods. Except for Text-only and Image-only, all other methods are provided with a reference image and a corresponding relative caption. This evaluation represents the core functionality offered by ZeroSight as a ZS-CIR dataset.

**Evaluation for CIR Methods.** We also provide an evaluation for CIR methods. We divide ZeroSight into training and test sets in a ratio of 9:1, with the two sets containing 49,266 and 5,474 queries respectively. Since the queries in ZeroSight have multiple ground truths, it is essential to create one training data for each reference image, its relative caption and each positive target image. During inference, CIR methods only need to be provided with the reference image and its relative caption. This evaluation broadens the applicability of ZeroSight, making it suitable for both ZS-CIR and CIR, thereby offering a richer selection for different CIR methods.

### 5.2 EVALUATION METRICS

**Mean Average Precision (mAP).** On ZeroSight, given that each query has multiple target images serving as ground truth, we utilize mAP, a more fine-grained metric to take into account the rank of retrieval results. Firstly, for each positive target image, we calculate $P_i^j@j(1 \le i \le a, 1 \le j \le k)$ based on their rank positions, which represents the precision when the $i$-th positive target image is at the $j$-th rank position. Here, $a$ is the number of positive target images. Next, we apply AP@k for

each query, where k represents the number of top-ranked retrieval results. Finally, we calculate the mean AP@k, as detailed in App. F.

**Positive-Negative Ranking mAP (PNR-mAP).** Traditional evaluation metrics like mAP only consider the distribution of positive target images in the retrieval results, ignoring the presence and ranking of negative target images. The ranking order of negative target images relative to positive ones can also indicate retrieval quality. To address this, we propose the positive-negative ranking mAP (PNR-mAP) metric. Similar to mAP, we first calculate $P_i^j@j$ for each positive target image. Then, for the $i$-th positive target image at the $j$-th position, we compute the positive-negative ranking weight $w_i$ as follows:

$$w_i = \begin{cases} \frac{\sum_{x=1}^{l} \frac{N_x}{j}}{l} & \text{if } l \geq 1, \\ 1 & \text{if } l = 0. \end{cases}, \tag{6}$$

where $l(0 \leq l < j)$ is the number of negative target images preceding the $i$-th positive target image at the $j$-th position, and $N_x(1 \leq N_x < j, 1 \leq x \leq l)$ denotes the rank position of the $x$-th negative target image. If more negative target images are ranked further ahead of a positive target image, its positive-negative ranking weight will be smaller. We then calculate the positive-negative ranking average precision at k by incorporating these weights, as follows:

$$PNR\text{–}AP@k = \frac{\sum_{i=1}^{b} w_i \times P_i^j@j}{\min(a, k)}, \tag{7}$$

Lastly, with PNR-AP@k, we obtain the Positive-Negative Ranking Mean Average Precision at k.

## 5.3 RESULTS

**Performance of ZS-CIR Methods on ZeroSight.** The evaluation results on ZeroSight, outlined in Table 2, provide a comprehensible comparison of ZS-CIR methods. Based on these results, we observed that: (1) For the mAP metric, with different CLIP backbones, image-based methods achieve the best results, while methods based on image-to-text, such as LDRE, perform worse. This indicates that ZeroSight places more emphasis on the information contained within the image itself. (2) For the PNR-mAP metric, the performance for all methods are worser than their corresponding mAP performance. This demonstrates that relying solely on image similarity is insufficient to correctly distinguish negative target images that closely resemble positive target images. (3) In addition, training-based ZS-CIR methods outperform training-free methods on all backbones, which shows that learning based on image features during training can significantly improve the performance on ZeroSight. (4) mAP and PNR-mAP are consistent in comparing method performance, but the average mAP of 19.94 across all methods is 22.93% higher than the PNR-mAP of 16.22. This indicates that mAP does not account for the ranking of negative target images, leading to inflated retrieval performance and exaggerating the capabilities of CIR methods. This also demonstrates that our benchmark presents a challenging task.

**Performance of CIR Methods on ZeroSight.** We evaluate the performance of different CIR methods on our test set after training on our training set, as shown in Table 3. Among them, CLIP4CIR achieves the best performance under the PNR-mAP metric, while APTEMIS performs the worst under the same metric.

Additionally, we compare the performance of these CIR methods on the FashionIQ validation set before and after training on our training set. Details are provided in the App. G.1.

**Performance of SC4CIR Method.** To validate the effectiveness of SC4CIR, we conduct experiments using SEIZE for training-free methods and LinCIR for training-based methods. The experimental results are highlighted in Table 2. Based on these results, we can make the following observations: (1) For mAP and PNR-mAP metrics, the average mAP of 25.47 across all methods represents a 5.90% improvement

Table 3: Performance of CIR Methods on the ZeroSight test set.

| Method | PNR-mAP@k | | | | |
|---|---|---|---|---|---|
| | k=5 | k=10 | k=25 | k=50 | Avg. |
| ARTEMIS(ICLR'2022) | 14.34 | 15.51 | 17.06 | 17.92 | 16.21 |
| AMC(TOMM'2023) | 14.91 | 16.44 | 18.47 | 19.47 | 17.32 |
| VAL(CVPR'2020) | 19.25 | 22.04 | 25.75 | 26.92 | 23.49 |
| TIRG(CVPR'2019) | 20.43 | 23.52 | 26.63 | 27.17 | 24.44 |
| MAAF(ArXiv'2020) | 20.16 | 24.72 | 28.33 | 29.84 | 25.76 |
| DCNet(AAAI'2021) | 25.31 | 26.01 | 26.90 | 27.92 | 26.54 |
| DWC(AAAI'2024) | 23.14 | 25.33 | 28.24 | 29.85 | 26.64 |
| CLIP4CIR(TOMM'2023) | **23.67** | **29.23** | **33.44** | **34.93** | **30.32** |

over the previous average mAP of 24.05 and the average PNR-mAP of 22.11 across all methods

is 12.86% higher than the previous average PNR-mAP of 19.59. All methods show improvement when combined with SC4CIR, indicating its effectiveness. (2) For the mAP metric, the average result of 37.54 in all training-based methods is 1.79% higher than the previous average result of 36.88 and the average result of 17.42 in training-free methods shows an improvement of 12.39% over the previous average results of 15.50. For the PNR-mAP metric, the average result of 34.49 across all training-based methods shows an improvement of 6.35% over the previous average result of 32.43 and the average result of 13.86 in training-free methods is 25.66% higher than the previous average results of 11.03. Compared to training-based methods, training-free methods show greater improvement when using SC4CIR. This indicates that SC4CIR can help training-free methods gain the capabilities provided by training. The results on common datasets are provided in the App. H.

## 5.4 ABLATION STUDY

In this section, we conduct ablation studies to evaluate the individual contributions of components in our method. We select the SEIZE method with a ViT-L backbone as the baseline to evaluate the impact of our SC4CIR on its performance on the CIRCO test set. The results of the ablation study are shown in Table 4.

**Impact of Reverse Process 1.** Compared to Reverse Process 2, it exhibits improvement but with a relatively limited margin. This is because Reverse Process 1, despite serving as a reverse process for verification, fundamentally relies on the original architecture of the method for CIR operations. It merely maximizes the utilization of inherent capabilities of the baseline method, thus its effectiveness remains constrained by the original architecture of the method and initial retrieval results.

Table 4: Ablation of SC4CIR on CIRCO test set.

| Method | mAP@k | | | | |
|---|---|---|---|---|---|
| | k=5 | k=10 | k=25 | k=50 | Avg. |
| SEIZE | 24.98 | 25.82 | 28.24 | 29.35 | 27.10 |
| SEIZE + Reverse Process 1 | 25.07 | 26.64 | 29.44 | 30.42 | 27.89 |
| SEIZE + Reverse Process 2 | 25.78 | 26.93 | 30.24 | 30.64 | 28.40 |
| SEIZE + SC4CIR | **26.87** | **27.93** | **31.28** | **32.03** | **29.53** |

**Impact of Reverse Process 2.** The results demonstrates significantly greater improvement over Reverse Process 1. This shows that: (1) Unlike Reverse Process 1, Reverse Process 2 bypasses the limitations of the original method by leveraging the full capacity of Large Vision-Language Models (LVLMs, e.g., GPT-4o), enabling superior performance unbound by the constraints of the baseline method. (2) The LVLM exploits complete visual information from both reference and target images to refine ranking, underscoring that the original method does not make the full use of visual cues in reference images. (3) This under-utilization of visual information further demonstrates the necessity of constructing a dataset that emphasizes the visual information contained within images.

## 5.5 CASE STUDY

We select three representative methods, Text-only, Image-only and LinCIR, to compare their top-5 retrieval results on our dataset, with details in the Figure 5. Images with red borders represent negative target images, while images with green borders represent positive target images. This comparison presents that our ZeroSight is a challenge to the existing ZS-CIR methods.

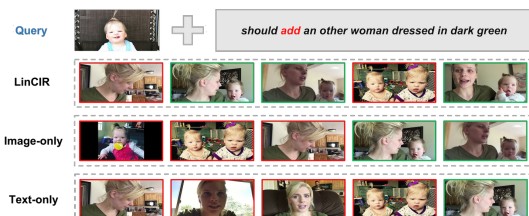

Figure 5: **Case study of our ZeroSight dataset.**

## 6 CONCLUSION

In this paper, we introduce ZeroSight, a novel benchmark for ZS-CIR that addresses the limitations of existing datasets by ensuring visual and semantic consistency through video-sourced data, ensuring a true zero-shot scenario by using data that has not been included in the pre-training of models like CLIP. Our multi-stage LLM-assisted pipeline generates high-quality queries with multiple positive and negative target images, providing a robust evaluation framework. Additionally, we introduce a novel, training-free MLLM-driven ZS-CIR method called SC4CIR, designed to identify hard negative targets in retrieval tasks. Experiments show that current CLIP-based CIR methods perform inflatedly on existing datasets, highlighting the need for ZeroSight. Our benchmark sets a new standard for both ZS-CIR and CIR research, fostering the development of more reliable and generalizable methods.

**Ethics Statement.** We have read and adhere to the ICLR Code of Ethics. Our research is conducted exclusively on publicly available video datasets and the ZS-CIR benchmark, and all datasets are used in accordance with their licenses. Our pipeline leverages large pre-trained models (e.g., OpenCLIP, ViT, BERT), which, like others of this type, may reflect limitations of their training data. While our method does not directly address such issues, it does not introduce additional risks. The intended use of ZeroSight is to construct a truly zero-shot benchmark, particularly suited for domains where the reference and target images for retrieval are highly similar (e.g., retrieval of medical images or analysis of remote sensing image retrieval). A positive ethical aspect is its contribution to innovation: it pioneers the use of video data sources to build a ZS-CIR benchmark, thereby minimizing the potential for data leakage to the greatest extent possible. We declare no competing interests.

**Reproducibility Statement.** We have made every effort to ensure the reproducibility of our work. The complete code for constructing ZeroSight and the full implementation of SC4CIR have been made publicly available under an anonymous repository. The code is accessible at `https://anonymous.4open.science/r/ZeroSight-CFE1`. All details regarding environment configuration, parameter settings, and model versions are provided in the code repository. We believe these measures will enable other researchers to reproduce our work and further advance the field.

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

# Appendix

## A  PROBLEM STATEMENT

**Zero-Shot Composed Image Retrieval (ZS-CIR).** The task of Composed Image Retrieval (CIR) can be defined as a multi-modal retrieval problem. Given a reference image $I^r$ and a relative caption $T$, the objective is to retrieve the target image $I^t$ from an image database $\mathcal{D}$ that aligns with the relative caption while preserving the underlying semantic content of the image that has not been explicitly mentioned. Zero-shot CIR further requires that no training samples are available, which thereby should be conducted with out-of-shelf tools.

## B  MORE RELATED WORK

**Multi-modal Large Language Models:** Building on the powerful language capabilities of Large Language Models (LLMs), such as GPT-4 and LLaMA, recent work has explored integrating multi-modal information, leading to MLLMs. Models like GPT-4v, GPT-4o, and Gemini pioneered unified architectures for diverse vision-language tasks beyond simple projection. These models are being applied in an expanding array of fields, where they exhibit outstanding performance. which are pre-trained to integrate visual information into LLM and are post-trained to align with users. OSrCIR exemplifies this advancement by leveraging MLLMs to jointly preserve visual semantics and linguistic intent through unified latent space modeling. While prior CIR methods often relied on fine-tuning specialized modules derived from such models, our research demonstrates that effectively combining

vision-language models with an LLM enables zero-shot CIR without additional training. Specifically, in this work, we leverage the MLLM's reasoning capability to infer the differences between reference and target images and to generate the reverse instruction based on the multi-modal query.

## C  BASELINES

To conduct a comprehensive comparison, we select the following state-of-the-art baselines for both ZS-CIR and CIR. We conducted the experiments by running these baselines on 8 × NVIDIA A800 GPUs.

**Baselines for ZS-CIR** *Captioning*: A method that employs the pre-trained captioning model BLIP-2 to generate captions of reference images and extract text features of the captions by CLIP text encoder to retrieve. *Text-only*: Only the relative caption features, extracted by the CLIP text encoder, are utilized as retrieval features to calculate similarity for retrieval. *Image-only*: Only the features of reference images, which are extracted by the CLIP image encoder, are used to compute similarity for retrieval. *CIReVL* Karthik et al. (2024): A training-free method using a pre-trained generative VLM and asking an LLM to recompose the caption based on the textual target modification for retrieval. *LDRE* Yang et al. (2024b): A training-free method utilizing out-of-shelf tools to accurately retrieve a composed image based on a reference image and a relative caption. *SEIZE* Yang et al. (2024a): A novel method based on LDRE without training. *PALAVRA* Cohen et al. (2022b): A two-stage training-dependent approach based textual inversion with a pre-trained mapping function and a subsequent optimization of the pseudo-word token. *SEARLE* Baldrati et al. (2023): A training-dependent method where pseudo-word tokens of unlabeled images are generated with an optimization-based textual inversion and then distill their knowledge to a textual inversion network. *SEARLE-OTI* Agnolucci et al. (2024): A variant of SEARLE without the distillation network. *Pic2Word* Saito et al. (2023): A training-dependent method employs a pre-trained textual inversion network optimized by contrastive loss to capture the pseudo-word token, which is combined with the relative caption for retrieval. *LinCIR* Gu et al. (2024): A language-only training framework for ZS-CIR.

**Baselines for CIR** *ARTEMIS* Delmas et al. (2022): A method based on two training jointly modules and their respective text-guided light-weight attention layers, where each handles one modality of the query. *AMC* Zhu et al. (2023): A method that leverages reinforcement learning to provide the model compression policy. *VAL* Chen et al. (2020a): A method fusing vision and language features via attention learning at varying representation depths. *TIRG* Vo et al. (2019): A method having the text modify the image feature via a gated residual connection. *MAAF* Dodds et al. (2020): A method using attention over undifferentiated text and image queries. *DCNet* Kim et al. (2021): A method that embeds the relationships between cells and their marker genes in the neural network, and can infer the cell landscape with more than 400 cell types based on bulk RNA-seq data. *DWC* Huang et al. (2024): A method to solve the inherent modality importance disparity, biased labeling noises, and modality gaps. *CLIP4CIR* Baldrati et al.: A two-stage method that combines task-oriented fine-tuning with the training of a Combiner network which can perform a fine-grained merging of the multimodal features.

## D  STATISTICAL ANALYSIS

### D.1  VIDEO CATEGORIES

Our dataset is collected from videos organized into 12 primary categories and 36 subcategories. These categories are depicted in Figure 6, illustrating the the diversity and inclusiveness of video types within our dataset.

### D.2  QUERY CATEGORIES

To facilitate a more comprehensive evaluation for ZS-CIR methods, we classify the queries into 6 distinct categories as shown in Figure 7. Each category corresponds to one specific assessment for ZS-CIR methods. The criteria for these categories are as follows: (1) **Addition**: Evaluating retrieval capability when adding elements. (2) **Subtraction**: Evaluating retrieval capability when removing elements. (3) **Background Change**: Evaluating retrieval capability when changing different

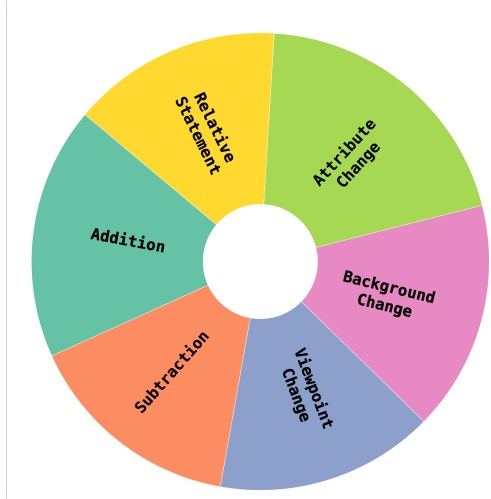

Figure 6: Video distribution · Figure 7: Query distribution

Table 5: Comparison results of different models before and after fine-tuning on ZeroSight using FashionIQ validation set. The best scores are highlighted in bold.

| Model | Stage | Shirt | | Dress | | Toptee | | Average | |
|---|---|---|---|---|---|---|---|---|---|
| | | R@10 | R@50 | R@10 | R@50 | R@10 | R@50 | R@10 | R@50 |
| MAAF(ArXiv'2020) | Pre-train | 19.68 | **35.92** | 15.32 | 34.75 | 21.01 | 41.36 | 18.67 | 37.34 |
| | Fine-tune | **20.07** | 35.38 | **16.81** | **35.60** | **21.98** | **42.78** | **19.62** | **37.92** |
| CLIP4CIR(TOMM'2023) | Pre-train | 25.32 | 41.46 | 20.92 | 40.70 | 27.49 | 47.53 | 24.58 | 43.23 |
| | Fine-tune | **26.84** | **45.14** | **21.47** | **43.43** | **28.35** | **49.72** | **25.55** | **46.10** |
| ARTEMIS(ICLR'2022) | Pre-train | 16.25 | **28.79** | 8.39 | 21.99 | 14.37 | **29.07** | 13.00 | **26.62** |
| | Fine-tune | **16.81** | 28.24 | **9.65** | **22.05** | **15.10** | 28.50 | **13.85** | 26.26 |
| AMC(TOMM'2023) | Pre-train | 15.66 | 26.97 | 8.57 | 20.84 | 13.96 | 27.23 | 12.73 | 25.01 |
| | Fine-tune | **16.19** | **27.16** | **9.31** | **21.22** | **14.54** | **27.41** | **13.35** | **25.26** |

backgrounds. (4) **Viewpoint Change**: Evaluating retrieval capability when shifting viewpoints. (5) **Attribute Change**: Evaluating retrieval capability when changing attributes of objects. (6) **Relative Statement**: Evaluating retrieval capability involving relative description of objects.

## E  QUALITY EVALUATION

We utilized three different LVLMs to construct queries following the same pipeline initially. The constructed queries were then manually assessed from four different aspects, as shown in Table 7. Based on the assessment, we selected GPT-4o for our pipeline.

## F  DETAILED EVALUATION METRICS

**Mean Average Precision (mAP).** On ZeroSight, given that each query has multiple target images serving as ground truth, we utilize mAP, a more fine-grained metric to take into account the rank of retrieval results. Firstly, for each positive target image, we calculate $P_i^j@j(1 \le i \le a, 1 \le j \le k)$ based on their rank positions as follows:

$$P_i^j@j = \frac{i}{j}, \tag{8}$$

Table 6: Comparison results of different models before and after fine-tuning on ZeroSight using CIRR test set. The best scores are highlighted in bold.

| Model | Stage | Recall@k | | | | | |
| | | k=1 | k=5 | k=10 | k=50 | k=100 | Avg. |
|---|---|---|---|---|---|---|---|
| MAAF(ArXiv'2020) | Pre-train | 20.99 | 31.04 | 49.21 | 62.27 | **87.42** | 50.18 |
| | Fine-tune | **22.53** | **32.87** | **49.95** | **63.28** | 86.58 | **51.04** |
| CLIP4CIR(TOMM'2023) | Pre-train | **6.17** | 22.53 | 44.58 | 59.74 | 86.48 | 43.90 |
| | Fine-tune | 5.83 | **25.30** | **49.86** | **65.52** | **89.76** | **47.25** |
| ARTEMIS(ICLR'2022) | Pre-train | 11.36 | 27.25 | 41.56 | 57.09 | 84.00 | 44.25 |
| | Fine-tune | **12.49** | **29.97** | **45.71** | **62.79** | **86.39** | **47.47** |

Table 7: Evaluation of LVLMs in constructing queries. **IEM**: Image Element Moderation. **MTIA**: Multiple Target Image Abundance. **TIC**: Text-Image Consistency. **SC**: Semantic Conciseness.

| LVLM | IEM | MTIA | TIC | SC | Average |
|---|---|---|---|---|---|
| Gemini 1.5 Pro | 3.80 | 3.31 | 4.48 | 4.56 | 4.04 |
| Qwen2.5-VL-72B | 3.62 | 2.64 | 4.29 | 4.52 | 3.78 |
| GPT-4o | **4.17** | **4.44** | **4.64** | **4.59** | **4.46** |

which represents the precision when the $i$-th positive target image is at the $j$-th rank position. Here, $a$ is the total number of positive target images. Then, we apply Average Precision at k (AP@k) for each query, where $k$ represents the number of top-ranked retrieval results, as shown below:

$$AP@k = \frac{\sum_{i=1}^{b} P_i^j @ j}{\min(a, k)},$$ (9)

where $b$ $(0 \leq b \leq k)$ is the number of positive target images among the top $k$ retrieval results for each query. Lastly, we calculate the mean AP@k as follows:

$$mAP@k = \frac{\sum_{q=1}^{n} AP_q @ k}{n},$$ (10)

where $AP_q@k$ represents the $AP@k$ of the retrieval results for the $q$-th query, and $n$ is the total number of queries.

**Positive-Negative Ranking mAP (PNR-mAP).** In the ZS-CIR task, negative target images refer to images that closely resemble positive target images but do not meet the relative caption to be classified as positive. Traditional evaluation metrics like mAP only consider the distribution of positive target images in the retrieval results, ignoring the presence and ranking of negative target images. The ranking order of negative target images relative to positive ones can also indicate retrieval quality. For example, two retrieval results with the same mAP value can differ significantly if, in one, negative target images are ranked before positive ones, while in the other, the opposite is true. The latter is clearly better, but mAP cannot distinguish between them. To address this, we propose the positive-negative ranking mAP (PNR-mAP) metric. Similar to mAP, we first calculate $P_i^j @ j$ for each positive target image. Then, for the $i$-th positive target image at the $j$-th position, we compute the positive-negative ranking weight $w_i$ as follows:

$$w_i = \begin{cases} \frac{\sum_{x=1}^{l} \frac{N_x}{j}}{l} & \text{if } l \geq 1, \\ 1 & \text{if } l = 0. \end{cases}$$ (11)

where $l(0 \leq l < j)$ is the number of negative target images preceding the $i$-th positive target image at the $j$-th position, and $N_x(1 \leq N_x < j, 1 \leq x \leq l)$ denotes the rank position of the $x$-th negative target image. If more negative target images are ranked further ahead of a positive target image, its positive-negative ranking weight will be smaller. We then calculate the positive-negative ranking average precision at k by incorporating these weights, as follows:

$$PNR\text{--}AP@k = \frac{\sum_{i=1}^{b} w_i \times P_i^j @ j}{\min(a, k)},$$ (12)

Lastly, with PNR-AP@k, we obtain the Positive-Negative Ranking Mean Average Precision at k, similar to the standard mAP@k calculation method, as shown in Eq. 10.

Table 8: Performance of Various Query Categories. **Add.**: Addition. **Sub.**: Subtraction. **View.**: Viewpoint Change. **Back.**: Background Change. **Att.**: Attribute Change. **Rel.**: Relative Statement. **Avg.**: Average.

| Method | Add. | Sub. | View. | Back. | Att. | Rel. | Avg. |
|---|---|---|---|---|---|---|---|
| TIRG(CVPR'2019) | 23.81 | 21.65 | 24.19 | 22.01 | 26.17 | 22.97 | 23.52 |
| VAL(CVPR'2020) | 22.38 | 20.22 | 22.68 | 20.62 | 24.54 | 21.51 | 22.04 |
| DCNet(AAAI'2021) | 26.23 | 24.05 | 26.73 | 24.35 | 28.91 | 25.44 | 26.01 |
| MAAF(ArXiv'2020) | 24.98 | 22.81 | 25.41 | 23.14 | 27.49 | 24.16 | 24.72 |
| ARTEMIS(ICLR'2022) | 16.04 | 13.94 | 16.02 | 14.48 | 17.36 | 15.04 | 15.51 |
| CLIP4CIR(TOMM'2023) | 29.35 | 27.15 | 30.01 | 27.38 | 32.45 | 28.63 | 29.23 |

## G    MORE EXPERIMENTS

### G.1    ZERO-SHOT RETRIEVAL PERFORMANCE FINE-TUNING ON ZEROSIGHT

**FashionIQ** Table 5 shows the performance of these CIR methods on the FashionIQ validation set before and after fine-tuning on our training set. Based on the results, we have following observations: (1) After fine-tuning on our dataset, all methods exhibit retrieval capabilities that are either comparable to or improved from their pre-training performance. This indicates that our dataset, as a ZS-CIR dataset, can provide training data for CIR methods. (2) Among the methods, CLIP4CIR shows the highest improvement after fine-tuning, with a 3.13% increase in average Recall@50. This further demonstrates the effectiveness of the training data provided by our dataset.

**CIRR** The experimental results, outlined in Table 6, show the performance of these CIR methods on the CIRR test set before and after fine-tuning on our training set. Based on the results, we can make the following observations: (1) After fine-tuning on our dataset, all methods also can exhibit retrieval capabilities that are either comparable to or improved from their pre-training performance. This indicates that our dataset, as a ZS-CIR dataset, can provide training data for CIR methods. (2) Among the methods, as the same as results on the FashionIQ validation set, CLIP4CIR shows the highest improvement after fine-tuning, with a 4.65% increase in average Recall@50. This further demonstrates the effectiveness of the training data provided by our dataset.

### G.2    PERFORMANCE OF VARIOUS QUERY CATEGORIES

The evaluation results, as shown in Table 8, demonstrate the performance of these CIR methods for six different categories of queries. Based on these results, we have the following observations: (1) For queries of the Attribute Change category, all methods perform relatively well, indicating that queries of the Attribute Change category are relatively simple for current CIR methods. (2) Conversely, for queries of the Background Change category, all methods have poor performances, suggesting that queries of the Background Change category in our dataset are challenging for current CIR methods. (3) Overall, all results of all methods across these six categories of queries are not very well, indicating that our dataset remains challenging for existing CIR methods.

## H    MODEL COMPARISON

### H.1    MULTI-TARGET PERFORMANCE ON CIRCO AND ZEROSIGHT

Since both CIRCO and ZeroSight are designed for one-to-many retrieval tasks and employ the same evaluation metric (mAP), we have also conducted experiments to compare the performance of different models on these two datasets, as shown in Table 9. The results of the I2T method on ZeroSight are generally lower than those on CIRCO. Specifically, the mAP@5 result of *SEIZE* on ZeroSight is 16.63, which is a significant decrease of 48.77% compared to the result of 32.46 on CIRCO. On the other hand, the results of other methods on ZeroSight are generally higher than those on CIRCO. For instance, the mAP@5 result of *LinCIR* on ZeroSight is 28.13, which is a substantial increase of 55.24% compared to the result of 12.59 on CIRCO. This also reflects that CIRCO fails to balance the information from the reference image and the relative caption, allowing

Table 9: Results of comparison among different models on CIRCO and ZeroSight test sets.

| Backbone | Method | Training-free | ZeroSight | | mAP@k | | | CIRCO | | mAP@k | | |
|----------|--------|---------------|-----------|------|-------|-------|------|-------|-------|-------|-------|------|
| | | | k=5 | k=10 | k=25 | k=50 | Avg. | k=5 | k=10 | k=25 | k=50 | Avg. |
| ViT-B/32 | CIReVL(ICLR'2024) | ✔ | 7.76 | 8.51 | 9.57 | 10.12 | 8.99 | 14.94 | 15.42 | 17.00 | 17.82 | 16.30 |
| | SEIZE(MM'2024) | ✔ | 10.64 | 11.61 | 12.94 | 13.65 | 12.21 | 19.04 | 19.64 | 21.55 | 22.49 | 20.68 |
| | PALAVRA(ECCV'2022) | ✗ | 12.67 | 13.89 | 15.51 | 16.36 | 14.61 | 4.61 | 5.32 | 6.33 | 6.80 | 5.77 |
| | SEARLE-OTI(ICCV'2023) | ✗ | 21.39 | 24.17 | 26.45 | 27.27 | 24.82 | 7.14 | 7.83 | 8.99 | 9.60 | 8.39 |
| | SEARLE(ICCV'2023) | ✗ | 22.65 | 25.58 | 27.98 | 28.84 | 26.26 | 9.35 | 9.94 | 11.13 | 11.84 | 10.57 |
| | SEIZE+SC4CIR(Ours) | ✔ | 12.59 | 13.71 | 15.21 | 16.12 | 14.41 | 22.16 | 22.89 | 24.08 | 24.59 | 23.43 |
| ViT-L/14 | Captioning | ✔ | 5.86 | 6.68 | 7.74 | 8.32 | 7.15 | 1.65 | 1.96 | 2.42 | 2.71 | 2.19 |
| | Text-only | ✔ | 8.38 | 9.19 | 10.27 | 10.84 | 9.67 | 2.63 | 2.85 | 3.30 | 3.58 | 3.09 |
| | CIReVL(ICLR'2024) | ✔ | 9.45 | 10.40 | 11.74 | 12.42 | 11.00 | 18.57 | 19.01 | 20.89 | 21.80 | 20.07 |
| | Image-only | ✔ | 15.62 | 17.26 | 19.38 | 20.54 | 18.20 | 1.28 | 1.70 | 2.35 | 2.69 | 2.01 |
| | SEIZE(MM'2024) | ✔ | 13.02 | 14.36 | 15.98 | 16.80 | 15.04 | 24.98 | 25.82 | 28.24 | 29.35 | 27.10 |
| | Pic2Word(CVPR'2023) | ✗ | 15.06 | 18.09 | 19.78 | 20.21 | 18.28 | 8.72 | 9.51 | 10.64 | 11.29 | 10.04 |
| | SEARLE-OTI(ICCV'2023) | ✗ | 24.93 | 30.68 | 33.89 | 34.71 | 31.05 | 10.18 | 11.03 | 12.72 | 13.67 | 11.90 |
| | SEARLE(ICCV'2023) | ✗ | 26.12 | 32.17 | 35.55 | 36.41 | 32.56 | 11.68 | 12.73 | 14.33 | 15.12 | 13.47 |
| | LinCIR(CVPR'2024) | ✗ | 28.13 | 34.48 | 38.03 | 38.93 | 34.89 | 12.59 | 13.58 | 15.00 | 15.85 | 14.26 |
| | SEIZE+SC4CIR(Ours) | ✔ | 15.13 | 16.56 | 18.03 | 18.91 | 17.16 | 26.87 | 27.93 | 31.28 | 32.03 | 29.53 |
| ViT-G/14 | CIReVL(ICLR'2024) | ✔ | 12.00 | 13.24 | 14.74 | 15.49 | 13.87 | 26.77 | 27.59 | 29.96 | 31.03 | 28.84 |
| | SEIZE(MM'2024) | ✔ | 16.63 | 18.37 | 20.47 | 21.54 | 19.25 | 32.46 | 33.77 | 36.46 | 37.55 | 35.06 |
| | Pic2Word(CVPR'2023) | ✗ | 16.07 | 19.39 | 21.25 | 21.73 | 19.61 | 5.54 | 5.59 | 6.68 | 7.12 | 6.23 |
| | SEARLE(ICCV'2023) | ✗ | 31.46 | 37.95 | 41.72 | 41.83 | 38.23 | 13.20 | 13.85 | 15.32 | 16.04 | 14.60 |
| | LinCIR(CVPR'2024) | ✗ | 32.00 | 38.57 | 42.38 | 42.51 | 38.87 | 19.71 | 21.01 | 23.13 | 24.18 | 22.01 |
| | SEIZE+SC4CIR(Ours) | ✔ | 18.11 | 19.76 | 21.93 | 22.98 | 20.70 | 34.48 | 36.81 | 38.93 | 39.88 | 37.53 |

the I2T methods to achieve good results even after losing some reference image information, which is logically unacceptable. Furthermore, the results of other CIR methods without this loss process are improved on ZeroSight compared to CIRCO, which also indicates that ZeroSight better balances the information from the reference image and the relative caption (reference-target image consistency), providing a fairer reflection of the effectiveness of different methods.

## H.2 PERFORMANCE ON CIRCO AND CIRR

**CIRCO:** The left section of Table 10 displays CIRCO test results. Based on them, we have the following observations: (1) Among the simpler baselines, *Image-only* and *Captioning* perform worse than *Text-only*, indicating the importance of relative text for CIR. Besides, *Captioning* outperforms *Image-only*, suggesting that textual features from image captions are more suitable for CIR than direct visual features. (2) Among I2T-based methods, the methods based on pre-trained captioning models, *CIReVL* and *SEIZE*, perform better than the methods based on pseudo-word, *PALAVRA*, *Pic2Word*, *SEARLE*, and *LinCIR*. This demonstrates that the captions generated by the captioning model possess semantics that are more suitable for CLIP text encoding compared to textual inversion. (3) In the absence of SC4CIR, *SEIZE* already outperforms all baselines across all metrics and CLIP backbones. When SC4CIR is applied, the performance improves even further. For instance, using ViT-L/14, *SEIZE with SC4CIR* outperforms *SEIZE* by 7.57% in mAP@5, 8.17% in mAP@10, 10.76% in mAP@25, and 9.13% in mAP@50, proving the effectiveness of SC4CIR for CIR methods.

**CIRR:** The right section of Table 10 presents CIRR test results. From these results, key observations include: (1) *Text-only* performs significantly better than *Image-only* and *Captioning*, indicating a minimal correlation between reference and target images due to the noisy dataset and the lesser information provided by reference images. (2) Despite the noise, SC4CIR can continuously improve the performance of *SEIZE* and *SEIZE with SC4CIR* outperforms all baselines across all CLIP backbones. This highlights the robustness and adaptability of SC4CIR even in noisy data and diverse scenarios. (3) Using ViT-L/14 CLIP, *SEIZE with SC4CIR* surpasses the second-best method *SEIZE* by 7.82% in Recall@1, 5.56% in Recall@5, and 3.80% in Recall@10, emphasizing the effectiveness of SC4CIR.

## H.3 PERFORMANCE ON FASHIONIQ

As shown in Table 11, we can have the following observations: (1) *SC4CIR* can consistently improve the performance of *LinCIR* across all three subsets, demonstrating the effectiveness of *SC4CIR* on

Table 10: Results of comparison among different models on CIRCO and CIRR test sets. Best scores are highlighted in bold.

| Backbone | Method | Training-free | CIRCO | | | | CIRR | | | | | |
| | | | mAP@k | | | | Recall@k | | | Rs@k | | |
| | | | k=5 | k=10 | k=25 | k=50 | k=1 | k=5 | k=10 | k=1 | k=2 | k=3 |
| ViT-B/32 | PALAVRA(ECCV'22) | ✗ | 4.61 | 5.32 | 6.33 | 6.80 | 16.62 | 43.49 | 58.51 | 41.61 | 65.30 | 80.94 |
| | SEARLE(ICCV'23) | ✗ | 9.35 | 9.94 | 11.13 | 11.84 | 24.00 | 53.42 | 66.82 | 54.89 | 76.60 | 88.19 |
| | SEARLE-OTI(ICCV'23) | ✗ | 7.14 | 7.83 | 8.99 | 9.60 | 24.27 | 53.25 | 66.10 | 54.10 | 75.81 | 87.33 |
| | CIReVL(ICLR'24) | ✔ | 14.94 | 15.42 | 17.00 | 17.82 | 23.94 | 52.51 | 66.00 | 60.17 | 80.05 | 90.19 |
| | SEIZE(MM'24) | ✔ | **19.04** | **19.64** | **21.55** | **22.49** | 27.47 | 57.42 | 70.17 | 65.59 | 84.48 | 92.77 |
| | SEIZE+SC4CIR(Ours) | ✔ | 22.16 | 22.89 | 24.08 | 24.59 | 29.58 | 59.39 | 73.16 | 68.36 | 85.61 | 93.48 |
| ViT-L/14 | Image-only | ✔ | 1.28 | 1.70 | 2.35 | 2.69 | 3.64 | 12.75 | 23.32 | 11.58 | 31.41 | 45.26 |
| | Text-only | ✔ | 2.63 | 2.85 | 3.30 | 3.58 | 20.51 | 43.21 | 55.08 | 60.39 | 80.02 | 90.05 |
| | Captioning | ✔ | 1.65 | 1.96 | 2.42 | 2.71 | 4.05 | 15.88 | 25.69 | 20.87 | 40.60 | 60.89 |
| | Pic2Word(CVPR'23) | ✗ | 8.72 | 9.51 | 10.64 | 11.29 | 23.90 | 51.70 | 65.30 | 53.76 | 74.46 | 87.08 |
| | SEARLE(ICCV'23) | ✗ | 11.68 | 12.73 | 14.33 | 15.12 | 24.24 | 52.48 | 66.29 | 53.76 | 75.01 | 88.19 |
| | SEARLE-OTI(ICCV'23) | ✗ | 10.18 | 11.03 | 12.72 | 13.67 | 24.87 | 52.31 | 66.29 | 53.80 | 74.31 | 86.94 |
| | LinCIR(CVPR'24) | ✗ | 12.59 | 13.58 | 15.00 | 15.85 | 25.04 | 53.25 | 66.68 | 57.11 | 77.37 | 88.89 |
| | CIReVL(ICLR'24) | ✔ | 18.57 | 19.01 | 20.89 | 21.80 | 24.55 | 52.31 | 64.92 | 59.54 | 79.88 | 89.69 |
| | SEIZE(MM'24) | ✔ | **24.98** | **25.82** | **28.24** | **29.35** | 28.65 | 57.16 | 69.23 | 66.22 | 84.05 | 92.34 |
| | SEIZE+SC4CIR(Ours) | ✔ | 26.87 | 27.93 | 31.28 | 32.03 | 30.89 | 60.34 | 71.86 | 68.43 | 87.31 | 94.99 |
| ViT-G/14 | Pic2Word(CVPR'23) | ✗ | 5.54 | 5.59 | 6.68 | 7.12 | 30.41 | 58.12 | 69.23 | 68.92 | 85.45 | 93.04 |
| | SEARLE(ICCV'23) | ✗ | 13.20 | 13.85 | 15.32 | 16.04 | 34.80 | 64.07 | 75.11 | 68.72 | 84.70 | 93.23 |
| | LinCIR(CVPR'24) | ✗ | 19.71 | 21.01 | 23.13 | 24.18 | 35.25 | 64.72 | 76.05 | 63.35 | 82.22 | 91.98 |
| | CIReVL(ICLR'24) | ✔ | 26.77 | 27.59 | 29.96 | 31.03 | 34.65 | 64.29 | 75.06 | 67.95 | 84.87 | 93.21 |
| | SEIZE(MM'24) | ✔ | **32.46** | **33.77** | **36.46** | **37.55** | 38.87 | 69.42 | 79.42 | 74.15 | 89.23 | 95.71 |
| | SEIZE+SC4CIR(Ours) | ✔ | 34.48 | 36.81 | 38.93 | 39.88 | 40.23 | 71.58 | 81.68 | 76.63 | 90.35 | 96.88 |

Fashion-IQ. (2) *LinCIR+SC4CIR* performs best on the R@10 metric of the Shirt subset and all metrics of the Toptee subset, even surpassing the performance of the trained CIR method *CASE* on these metrics. Additionally, *LinCIR+SC4CIR*'s performance on the R@50 metric of the Shirt subset and all metrics of the Dress subset is second only to *CASE*, proving that *SC4CIR* can consistently enhance the effectiveness of CIR methods and even help ZS-CIR methods achieve performance comparable to trained CIR methods to some extent.

Table 11: Results of comparison among different models on FashinIQ.

| Methods | Shirt | | Dress | | Toptee | | Average | |
| | R@10 | R@50 | R@10 | R@50 | R@10 | R@50 | R@10 | R@50 |
| CASE | 48.48 | 70.23 | 47.44 | 69.36 | 50.18 | 72.24 | 48.79 | 70.68 |
| TransAgg(Laion-CIR-Combined) | - | - | - | - | - | 34.36 | 55.13 | - |
| CoVR | - | - | - | - | - | 27.70 | 44.63 | - |
| LinCIR(ViT-G/14) | 46.76 | 65.11 | 38.08 | 60.88 | 50.48 | 71.09 | 45.11 | 65.69 |
| LinCIR(ViT-G/14)+SC4CIR | 49.34 | 68.02 | 41.53 | 63.91 | 53.18 | 74.33 | 48.02 | 68.75 |

## I  ANALYSIS OF SC4CIR EFFICIENCY

Firstly, even when scaling up existing methods at the same time cost, the performance of these methods may not necessarily improve. This is because the parameter settings given by existing methods are typically the result of numerous experiments by their authors, aiming to achieve a good balance between performance and cost.

Taking *SEIZE* (Semantic Editing Increment Benefits Zero-Shot Composed Image Retrieval) as an example, the left curve in Figure 5 of the paper shows the impact of the number of multiple captions on the performance of *SEIZE* on CIRCO, which can be converted into a table as Table 12, where N is the number of multiple captions. As shown in the table, when N increases from 10 to 14, the

performance of *SEIZE* shows a slight improvement. However, when N further increases to 20, the performance of *SEIZE* does not improve significantly. Increasing N from 10 to 20 essentially doubles the size of one of modules of *SEIZE*, but the effect is not as good as directly using *SC4CIR* when $N = 10$. The mAP@25 result of 31.28 for SC4CIR is even $11.4\%$ higher compared to the result of 28.1 for $N = 20$. This reflects that simply expanding the size of *SEIZE* does not necessarily improve its performance, and it will be better to combine it with *SC4CIR*.

Secondly, while $SC4CIR$ increases inference time, it provides stable and significant performance gains, evidenced by improvements of $29.39\%$ (*SEIZE*) and $6.86\%$ (*LinCIR*) in Avg PNR-mAP. Crucially, *SC4CIR* is a plug-and-play module, distinct in design and function from existing methods. It should therefore be evaluated on its ability to enhance state-of-the-art techniques, not solely on raw efficiency. Furthermore, its parameter N offers flexibility: users can tune it to balance effectiveness and computational cost based on specific application requirements.

Table 12: Performance Comparison of SEIZE Varying Numbers of Captions and SEIZE Integrated with SC4CIR on the CIRCO Test Set.

| Method | mAP@k | | | | |
|---|---|---|---|---|---|
| | k=5 | k=10 | k=25 | k=50 | Avg. |
| N=5 | 24.27 | 25.04 | 27.54 | 28.59 | 26.36 |
| N=10 | 24.59 | 25.14 | 27.97 | 29.03 | 26.68 |
| N=14 | 25.11 | 25.95 | 28.10 | 29.12 | 27.07 |
| N=20 | 24.95 | 25.92 | 28.12 | 29.16 | 27.04 |
| N=10&&+SC4CIR | 26.89 | 27.93 | 31.28 | 32.03 | 29.53 |

## J    CATEGORIZATION OF THE NOISE RATIO IN THE EXISTING BENCHMARKS

We randomly selected 200 pairs of queries from the validation sets of the CIRCO, CIRR, and FashionIQ datasets, and manually reviewed the number of content-inconsistent and reference-irrelevant queries in each dataset's corresponding 200 pairs. There are some important notes: (1) To ensure a fair comparison, we adopted a relaxed content-consistency criterion (requiring only core elements to match, acknowledging inherent subjectivity) rather than the stricter ZeroSight definition (which demands exact matches, e.g., from the same video/person). Three annotators independently labeled each query, and results reflect their average scores. (2) For CIRCO, queries targeting multiple images were counted as separate cases. (3) For FashionIQ, we sampled evenly across the Shirt, Dress, and Toptee subcategories.

The results are shown in Table 13, which reveals that FashionIQ exhibits the highest proportion of both noise types, with content inconsistency reaching $58.0\%$ and reference irrelevance at $17.0\%$. This is largely attributable to the practical impossibility of maintaining identical models across images in this domain. Furthermore, CIRCO and CIRR also show significant noise levels, with the combined proportion of these issues approaching one-third of queries. These findings demonstrate that content-inconsistent and reference-irrelevant noise is prevalent in existing benchmarks. While this initial analysis is based on a sample of 200 queries per dataset, we plan to extend this verification to the full datasets and will open-source the identified noisy case IDs to facilitate future dataset refinement efforts.

## K    APPLICATION OF ZEROSIGHT AND SC4CIR IN REAL-WORLD

As far as we know, in the medical field, when retrieving medical images, and in remote sensing image analysis tasks for urban planning and environmental monitoring, the target image obtained by modifying the reference image according to the relative caption must be very similar to the original reference image in the unmodified areas. This ensures the usability of the retrieval results. In these applications, the differences between images can be very subtle, yet the accuracy requirements for retrieval are very high. The practicality of our dataset in the real world lies in these applications. In correspondence with ZeroSight, *SC4CIR* is also better suited for applications that demand extremely high retrieval accuracy. In these applications, errors in retrieval results can lead to very serious consequences, so the importance of retrieval accuracy far outweighs the real-time nature of retrieval.

## L    NEGATIVE IMPACT OF PRE-TRAINED CLIP ON ZS-CIR

Given that the image pairs in the current ZS-CIR datasets are derived from image datasets that may have been pre-trained by CLIP, resulting in a spurious zero-shot scenario, it is crucial to demonstrate that this spurious zero-shot scenario can adversely affect

Table 13: Categorization of The Noise Ratio in CIRCO, CIRR and FashionIQ

| Benchmarks | Content-Inconsistent | Reference-Irrelevant |
|---|---|---|
| CIRCO | 26.66% | 6% |
| CIRR | 32.33% | 8% |
| FashionIQ | 58.00% | 17% |

ZS-CIR performance. We design an experiment to compare the performance of CLIP on the ZS-CIR task before and after training on ZeroSight image data. Since ZeroSight image data originates from video datasets post-2022, performing ZS-CIR directly on it will not introduce the influence of pre-training. Subsequently, we conduct contrastive learning training on the ZeroSight image data using image-caption pairs (captions generated by BLIP-2 Li et al. (2023)). The performance improvement in ZS-CIR following this training indicates the impact of CLIP pre-training. We consistently employ LDRE Yang et al. (2024b)

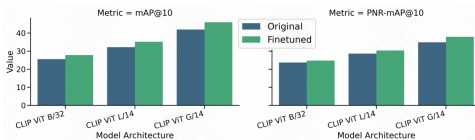

Figure 8: Performance difference in ZS-CIR before and after training on ZeroSight image data. The results demonstrate that pre-training on the same dataset inflates ZS-CIR performance, with larger CLIP models exhibiting more significant improvements.

for the CIR experiments. The experimental results, presented in Figure 8, demonstrate that for all CLIP architectures, pre-training on a specific image dataset leads to inflated results when performing composed image retrieval on ZS-CIR datasets constructed from the same dataset. Furthermore, the larger the CLIP model, the more pronounced the inflation; for instance, CLIP ViT-G/14 exhibited a 8.68% improvement.

## M  STATEMENT ON THE USE OF LARGE LANGUAGE MODELS

In line with the conference policy, we disclose that Large Language Models (LLMs) were utilized solely during the construction process of ZeroSight and as an auxiliary tool in the writing process. During the construction pipeline of ZeroSight, LLMs were employed only in the following procedures: (1) constructing the candidate reference image set, (2) generating candidate relative captions and final relative captions, and (3) evaluating the performance of different LVLMs when selecting the model for constructing ZeroSight. Detailed information can be found in Sec.3.2.

Furthermore, LLMs were used solely as writing assistants, with their involvement limited to improving grammar, refining sentence structure, and enhancing readability. All scientific contributions, including the development of ideas, methodology, experiments, and conclusions, were made exclusively by the authors, who take full responsibility for the content of this paper.

## N  VISUALIZATION

Figure 9 shows examples of our dataset, which contain queries with corresponding reference image, relative caption, positive target images and negative target images. In addition, to ensure a rich variety of queries in our dataset, we further divide all queries into six categories, including Addition, Subtraction, Viewpoint Change, Background Change, Attribute Change and Relative Statement. For example, if a relative caption contains the content "should add another woman dressed in dark green," the corresponding query should be classified as category "Addition." The reason is that the relative caption intends to add new content to its corresponding reference image and then perform retrieval based on this compose content. Furthermore, in this relative caption, the word "add" also reflects the core characteristic of category "Addition."

## O  PROMPTS

Below are the prompts used for selection of candidate reference images, and generation of candidate relative captions and final relative captions within our dataset by both the LVLM(such as GPT-4o) and the LLM(such as GPT-4) mentioned.

Figure 9: Examples of our ZS-CIR dataset. We divide all queries into six categories, including Addition, Subtraction, Viewpoint Change, Background Change, Attribute Change and Relative Statement. The words in relative caption, highlighted in red, indicate the core characteristic of the category to which the query belongs.

---

**Prompt for selecting candidate reference images(when selecting the first candidate reference image)**

Task Description:
You will be given several images extracted from video frames in sequence. You need to select one image that meets the following requirements.
Requirements:
1. The selected image must be aesthetically pleasing, conforming to human aesthetics, and must be clear without any blurry areas.
2. The selected image should be sufficiently distinct from the other images, and the difference should be describable in one sentence.
3. The number of people and objects in the image should be moderate, neither too many nor too few.
4. The overall arrangement of elements in the image should not appear too monotonous or too chaotic.
5. Images with only English words are not allowed to be selected.
6. Only use Arabic numerals to output the sequence number of the selected image, and do not output any other characters.
7. If there is no suitable image, output the number 0 using Arabic numerals, and do not output any other characters.
{Input image sequence}

**Prompt for selecting candidate reference images (when selecting other candidate reference images)**

Task Description:
You will be given several images extracted from video frames in sequence. The first image will be used as a reference image, and you need to select one image from the remaining images that meets the following requirements.
Requirements:
1. The selected image must not resemble the reference image and must have sufficient differences from it.
2. The selected image must be aesthetically pleasing, conforming to human aesthetics, and must be clear without any blurry areas.
3. The selected image should be sufficiently distinct from the other images (excluding the reference image), and the difference should be describable in one sentence.
4. The number of people and objects in the image should be moderate, neither too many nor too few.
5. The overall arrangement of elements in the image should not appear too monotonous or too chaotic.
6. Images with only English words are not allowed to be selected.
7. Only use Arabic numerals to output the sequence number of the selected image, ranging from 2 to 10, and do not output any other characters.
8. If there is no suitable image, output the number 1 using Arabic numerals, and do not output any other characters.
{Input image sequence}

**Prompt for generating candidate relative captions**

Task Description:
You will be given two images in sequence. You need to generate a declarative sentence that, when combined with the content of the first image, will enable a search engine to accurately retrieve the second image.
Output Example:
"should add more people in a bright room."
Requirement:
1. The output sentence should describe the modifications needed to change the first image into the second image.
2. The subject of the output sentence should be "the first image" and the predicate can be a series of verbs such as "increase", "enlarge", "reduce", "show", "zoom", etc. However, the output must follow the given output example, and the subject must be omitted.
3. Only this sentence can be output, and no other characters are allowed.
4. The declarative sentence must be output in English.
5. The modifications described in the declarative sentence should focus more on the elements within the images (people, objects, colors, numbers, environments, etc.).
{Input Image Sequence}

**Prompt for generating final relative captions**

Given several declarative sentences without subjects but with similar meanings, you need to generate a declarative sentence in the same format.

However, the generated declarative sentence should be able to summarize all the given declarative sentences.

I will start with "Given declarative sentences:" to provide you with the declarative sentences. When generating, strictly follow the given example, and do not generate any other characters. The generated summary declarative sentence must be simple enough and generated in one line.

Input example:

["shows a person standing alone in a room with patterned wallpaper and no visible candles.",
"shows a single person standing in front of a patterned wall.",
"shows a lone individual standing indoors with a different background and lighting.",
"shows a different woman in a dimly lit area with fewer visible light sources.",
"shows a person alone in dim lighting."]

Generated example:

"shows a person standing alone."

{Given declarative sentences}

