# OpenReview forum: "Never Seen Before: Benchmarking Genuine Zero-Shot Composed Image Retrieval with Consistent Video-Sourced Datasets"
_ICLR.cc/2026/Conference — Submitted to ICLR 2026_

### Official Review · Reviewer_Meo7 · 2025-10-15

**Soundness:** 3
**Presentation:** 1
**Contribution:** 3
**Rating:** 6
**Confidence:** 5

**Summary:**

This paper introduces ZeroSight, a new benchmark for Zero-Shot Composed Image Retrieval (ZS-CIR) that addresses critical flaws in prior datasets, namely data inconsistency and the violation of the true zero-shot assumption. ZeroSight ensures data consistency by using frames from the same video and guarantees a true zero-shot setting by using content published after CLIP's training cut-off.

The authors also propose SC4CIR, a training-free, MLLM-based module that improves retrieval by identifying hard negatives through symmetric consistency checks. A central finding is that previous benchmarks have significantly inflated the performance of existing CIR methods.

**Strengths:**

1. The motivation for establishing the ZeroSight benchmark is well-founded. It addresses a significant validity concern in the field by creating an evaluation set that is verifiably excluded from the pre-training data of common foundation models like CLIP, thereby enabling a more accurate assessment of true zero-shot CIR capabilities.
2. The paper proposes a ZS-CIR method, SC4CIR, and the accompanying experiments validate its effectiveness in improving retrieval performance on the new benchmark.
3. The scope of the work is substantial, encompassing the entire pipeline for dataset construction, the proposal of a new method, and an extensive experimental evaluation across a large number of existing methods.

**Weaknesses:**

1. Insufficient Contextualization of the Problem Domain: While the paper makes a valuable contribution to ZS-CIR, its impact could be significantly strengthened by a more comprehensive discussion of the research landscape. The manuscript does not fully articulate the unique challenges inherent to ZS-CIR, particularly in contrast to closely related tasks like Multi-turn CIR [1] and Composed Video Retrieval [2]. A clearer exposition of what makes ZS-CIR a distinct and difficult problem would better motivate the work and highlight the novelty of the proposed solutions. In addition, recent works in the CIR field can also be explored at the methodological level to highlight the contributions of SC4CIR.
2. Readability and Presentation: A primary concern that currently limits the paper's accessibility is its overall readability. Several key figures and illustrations, for instance, use a font size that is too small, which hinders comprehension of important details. Improving the clarity of the writing and ensuring all visual elements are legible would make the paper's technical contributions much easier for the reader to appreciate.

Addressing these issues would substantially enhance the paper's quality and clarity. I would be willing to reconsider my evaluation should these concerns be adequately resolved.


[1] Chen Y, Yang Z, Xu J, et al. MAI: A Multi-turn Aggregation-Iteration Model for Composed Image Retrieval[C]//The Thirteenth International Conference on Learning Representations. 2025.
[2] Thawakar O, Naseer M, Anwer R M, et al. Composed video retrieval via enriched context and discriminative embeddings[C]//Proceedings of the IEEE/CVF Conference on Computer Vision and Pattern Recognition. 2024: 26896-26906.

**Questions:**

1. The paper's investigation is centered on the image domain. Given the increasing research focus on more dynamic modalities, such as in zero-shot composed video retrieval, what is the specific motivation for addressing this problem in the static image domain? The authors should elaborate on the unique challenges of ZS-CIR that are not present or are fundamentally different from those in composed video retrieval.
2. To better situate this work's contribution, a discussion and comparison with related tasks in the video domain would be beneficial. Why were existing works on composed video retrieval not referenced or compared against, even as a point of discussion to delineate the scope?

[1] Uesugi K, Saito N, Maeda K, et al. Zero-shot Composed Video Retrieval with Projection Module Bridging Modality Gap[C]//2024 IEEE 13th Global Conference on Consumer Electronics (GCCE). IEEE, 2024: 6-7.

---

### Official Review · Reviewer_rWUG · 2025-10-31

**Soundness:** 2
**Presentation:** 3
**Contribution:** 3
**Rating:** 2
**Confidence:** 4

**Summary:**

This paper introduces ZeroSight, a novel benchmark for Zero-Shot Composed Image Retrieval (ZS-CIR) that addresses two critical limitations of existing datasets: (1) visual and semantic inconsistency between reference and target images, and (2) spurious zero-shot scenarios caused by using images from datasets that CLIP was trained on.

The authors construct the dataset by extracting frames from videos published after March 31, 2022, ensuring they were not included in CLIP's pre-training data. Through a multi-stage LLM-assisted pipeline involving 30+ annotators over 3 months, they create 197,313 candidate images and 54,740 queries, with each query containing an average of 5.16 positive and 10.89 hard negative targets.

Additionally, they propose SC4CIR (Symmetric Consistency for CIR), a training-free MLLM-driven method that identifies hard negatives through forward retrieval and two reverse processes. They also introduce PNR-mAP, a new evaluation metric that considers the ranking of negative target images. Extensive experiments on 27 methods demonstrate that existing ZS-CIR datasets inflate retrieval performance, while SC4CIR consistently improves performance across various methods, particularly achieving 25.66% improvement on training-free methods for PNR-mAP.

**Strengths:**

1. **Clear problem identification with quantitative evidence**: The paper identifies two fundamental issues in existing ZS-CIR datasets (image pair inconsistency and spurious zero-shot scenarios) and provides strong empirical support. Notably, Appendix J quantifies noise ratios in existing datasets (26-58% content-inconsistent pairs), and Appendix L demonstrates that CLIP pre-training on test data inflates performance by up to 8.68%. The use of video frames to ensure consistency through narrative continuity is a creative and principled solution.

2. **Comprehensive dataset with novel characteristics**: ZeroSight is substantially larger than existing datasets (197K images, 54K queries) and is the first to explicitly include hard negative targets (avg. 10.89 per query). The 6-category query classification (Addition, Subtraction, Viewpoint Change, Background Change, Attribute Change, Relative Statement) enables fine-grained analysis of method capabilities. The multi-stage construction pipeline involving LLM/MLLM assistance demonstrates significant engineering effort.

3. **Effective and well-validated contributions**: SC4CIR's symmetric consistency checking through forward and two reverse processes is intuitive and demonstrates consistent improvements across all 27 tested methods (5.90% mAP, 12.86% PNR-mAP average improvement). The PNR-mAP metric thoughtfully addresses the limitation of standard mAP by considering hard negative rankings. Experiments are comprehensive, covering multiple CLIP backbones, both training-free and training-based methods, and include ablation studies validating individual component contributions.

**Weaknesses:**

1. **Insufficient guarantees for true zero-shot scenario**: The reliance solely on publication dates (post-March 31, 2022) to exclude CLIP training data is fundamentally weak. YouTube videos can be re-uploaded, and similar content may exist across different videos. The paper lacks additional verification mechanisms such as perceptual hashing or CLIP embedding similarity checks against LAION-5B. Furthermore, the 8.68% improvement shown in Appendix L when fine-tuning CLIP on ZeroSight could be confounded by domain adaptation effects (video frames as a specific domain) rather than pure data exposure, making it difficult to definitively conclude this represents "inflated performance" in existing datasets.

2. **Overly complex pipeline without adequate justification**: The dataset construction involves numerous filtering stages with many hyperparameters (α₁=0.50, β₁=0.80, α₂=0.35, α₃=0.50, β₂=0.65, β₃=0.75, θ=0.85), yet lacks ablation studies demonstrating the necessity of each stage. The caption generation process is particularly convoluted: MLLM candidates → BERT similarity → complex filtering via Equation (3) → LLM aggregation. Equation (3) defining k(x) lacks intuitive explanation, and no comparison with simpler alternatives (majority voting, clustering) is provided. The manual review process ("manually review and select appropriate queries") is vaguely described without specifying rejection rates, selection criteria, or inter-annotator agreement, raising reproducibility and bias concerns.

3. **Incomplete evaluation and analysis of proposed method**: SC4CIR's design choices lack thorough justification: (1) N=30 for top candidates appears arbitrary without sensitivity analysis, (2) simple summation S = S₁ + S₂ + S₃ ignores potential scale differences and varying importance of similarity scores without exploring weighted combinations, (3) Reverse Process 1's accuracy in generating inverse text T' (e.g., "add woman" → "remove woman") is not validated with error case analysis, (4) computational cost of calling GPT-4o twice for each of 30 candidates per query is not discussed. The ablation study (Table 4) is only conducted on CIRCO, not on ZeroSight itself. The train/test split methodology (video-level vs. query-level) is not specified, creating potential data leakage concerns. PNR-mAP's weight function w_i lacks comparison with alternative designs (exponential decay, rank-based weighting), and its applicability beyond ZeroSight is unexplored.

**Questions:**

1. **Zero-shot verification**: Beyond publication dates, what additional mechanisms did you use to verify CLIP training data exclusion? Have you computed perceptual similarity or CLIP embedding distances between ZeroSight images and LAION-5B? Could the 8.68% improvement in Appendix L be attributed to domain adaptation rather than data exposure?

2. **Pipeline simplification and ablation**: (a) How much does performance degrade when replacing the complex caption generation (Equations 3-5) with simpler methods like majority voting or clustering? (b) What is the impact of removing individual filtering stages or changing the order of ViT/CLIP filtering? (c) How were all threshold values determined, and how sensitive is performance to these choices?

3. **SC4CIR design and cost**: (a) How does performance and computational cost vary with different N values (10, 50, 100)? (b) Does weighted combination (w₁S₁ + w₂S₂ + w₃S₃) improve over simple summation? (c) What is the accuracy of reverse text generation T' in Reverse Process 1, and what are common failure cases? (d) What are the actual wall-clock time and API costs for SC4CIR per query, and how does this trade off with performance gains? (e) Can you provide ablation results on ZeroSight (not just CIRCO)?

4. **Experimental details**: Is the train/test split performed at the video level or query level? If query-level, how do you prevent data leakage from queries of the same video? What percentage of queries were rejected during manual review, and what were the selection criteria?

5. **Metric and generalization**: Have you explored alternative weight functions for PNR-mAP? What are the mAP vs. PNR-mAP gaps on CIRCO, CIRR, and FashionIQ to validate whether the 22.93% gap is specific to ZeroSight or a general phenomenon?

---

### Official Review · Reviewer_dwp7 · 2025-11-01

**Soundness:** 3
**Presentation:** 2
**Contribution:** 2
**Rating:** 6
**Confidence:** 3

**Summary:**

The paper aims to build a more reliable benchmark for composed image retrieval (CIR) that ensures (1) visually and semantically consistent reference–target image pairs and (2) a truly zero-shot evaluation setting. To this end, the authors propose ZeroSight, a novel benchmark derived from video-sourced datasets that provides consistent reference–target pairs, multiple positive samples, and hard negatives per query. To address the challenges posed by this benchmark, the paper introduces SC4CIR (Symmetric Consistency for CIR); a training-free, MLLM-driven method designed to effectively identify hard negative targets through three symmetric consistency checks. The results show that existing CLIP-based models may have overestimated performance under previous benchmarks.

**Strengths:**

1. The motivation of the paper is solid. As the authors point out, current CIR evaluation benchmarks are relatively coarse. Incorporating partial edits under consistent reference and target pairs is needed for genuinely evaluating CIR performance. This research direction is both needed for this community.

2. The proposed pipeline that leverages video datasets to obtain consistent reference–target image pairs is well-motivated, and the overall generation process is clearly structured. Moreover, maintaining a true zero-shot setting adds further credibility and robustness to the proposed benchmark.

3. The proposed training-free method is simple yet effective, demonstrating strong performance under the newly established benchmark.

**Weaknesses:**

1. To claim that “existing CLIP-based models may have overestimated performance under previous benchmarks,” more detailed analyses comparing the proposed and prior benchmarks are needed. Currently, the paper only reports results on the newly proposed benchmark. Furthermore, even within the proposed benchmark, CLIP-based methods (SEARLE, ...) still perform better compared to the proposed approach. Additional comparative results and analyses on both benchmarks would strengthen this claim.

2. Although the proposed method is training-free, the additional generation processes involved may introduce substantial computational overhead. A more detailed analysis of the computational cost would be beneficial. Additionally, it would be valuable to evaluate the proposed method on existing benchmarks (CIRR, CIRCO, FashionIQ) to assess its generalizability beyond the proposed dataset.

3. The definition and motivation for the Positive–Negative Ranking mAP metric are unclear. Specifically, it is not evident how hard negatives are ranked within this formulation.

4. The paper’s writing can be significantly improved for clarity and readability. The notations, while formal, are overly complex ( St_i, Tt_i, Ts_i, St, ... ) and sometimes redundant, particularly in Section 3.2. In my personal opinion, complex mathematical notations would not be necessary for explaining the generation procedure.  Simplifying the mathematical notation and refining explanations would make the paper easier to follow.

I believe the benchmark itself is meaningful, but the explanation and metrics, and analyses might not be sufficient in the current stage, but I think it would be improved in the rebuttal phase.

**Questions:**

Wrote above

---

### Official Review · Reviewer_oJxf · 2025-11-04

**Soundness:** 3
**Presentation:** 3
**Contribution:** 2
**Rating:** 2
**Confidence:** 4

**Summary:**

This paper proposes a benchmark for zero-shot composed image retrieval (ZS_CIR), including a dataset with consistent reference-target paris from videos, a data construction pipeline, and evaluation methods ranking multiple positive and negative target images.

**Strengths:**

1. The paper is well-written and easy to follow.
2. The paper proposes a comprehensive data collection pipeline for the new dataset ZeroSight.

**Weaknesses:**

1.  **Limited novlety of the SC4CIR method**. SC4CIR heavily relies on existing LLM (GPT-4o), without introducing new retrieval architecture or learning paradigm. It is likely that the performance improvements are inherited from the multi-modal reasoning abilities of GPT-4o, instead of the proposed framework itself.
2. **Lack of efficiency analysis**. Efficiency evaluations (memory usage, GFLOPs, inference time, etc) should be provided to investigate the trade-off between model performance and computational cost.

**Questions:**

1. The technical novelte of the proposed SC4CIR method should be further clarified.
2. Efficiency evaluations should be provided to investigate the trade-off between model performance and computational cost.
3. Why using video data published after March 31, 2022 ensures zero-shot scenario? What if the video content and CLIP's pretraining data overlap?

---

### Meta-Review · Area_Chair_6wTw · 2026-01-05

**Summary:**

This paper introduces ZeroSight, a new benchmark for zero-shot composed image retrieval (ZS-CIR), constructed from video data to enforce reference–target consistency and a stricter zero-shot setting. It also proposes a training-free MLLM-based reranking method, SC4CIR, and a new evaluation metric (PNR-mAP).

Reviewers oJxf and rWUG raised concerns about the novelty and rigor of the method, noting that SC4CIR largely relies on existing MLLMs (e.g., GPT-4o) without introducing new modeling ideas, and that the zero-shot guarantee based solely on video publication dates is not fully convincing. Both reviewers also pointed out missing analysis of efficiency, cost, and sensitivity to design choices, which is particularly important given the heavy use of large models.

Reviewers dwp7 and Meo7 were more positive about the benchmark itself, agreeing that ZeroSight addresses real limitations of existing ZS-CIR datasets and is well motivated. However, even these reviewers noted that key claims (e.g., inflated performance of prior benchmarks) are not sufficiently supported by cross-benchmark comparisons, and that the pipeline, metrics, and explanations remain overly complex and under-analyzed in their current form.

Overall, while there is agreement that the benchmark direction is meaningful and the engineering effort is substantial, the lack of rebuttal leaves several core concerns unaddressed—particularly around novelty, validation of the zero-shot assumption, computational cost, and clarity of evaluation. Given the mixed scores (6/2/6/2) and the absence of responses to reviewer questions, I recommend rejection at this time.

**Reviewer Concerns:**

Since there was no rebuttal, the reviewers' concerns would remain.

**Reviewer Scores:**

Since there was no rebuttal, the reviewers' concerns would remain.

---

### Decision · Program_Chairs · 2026-01-26

Reject